# Effect of Foot Reflexology on Muscle Electrical Activity, Pressure, Plantar Distribution, and Body Sway in Patients with Type 2 Diabetes Mellitus: A Pilot Randomized Controlled Trial

**DOI:** 10.3390/ijerph192114547

**Published:** 2022-11-05

**Authors:** Thais Gebin Toledo, Larissa Alves Moreira Freire, Luciana Maria dos Reis, Andréia Maria Silva Vilela Terra, Adriana Teresa Silva Santos

**Affiliations:** 1Human Performance Research Laboratory, Institute of Motor Sciences, Federal University of Alfenas, Santa Clara Campus, Alfenas 37133-840, MG, Brazil; 2Post-Graduation in Rehabilitation Sciencies, Human Performance Research Laboratory, Institute of Motor Sciences, Federal University of Alfenas, Santa Clara Campus, Alfenas 37133-840, MG, Brazil

**Keywords:** type 2 diabetes mellitus, foot reflexology, rehabilitation, EMG, integrative and complementary medicine, balance

## Abstract

Objective: To verify the effect of foot reflexology on the electrical muscle activity of the lateral and medial gastrocnemius muscle, and to examine the distribution, plantar pressure, and body sway in patients with type 2 diabetes mellitus. Methods: This pilot randomized controlled trial enrolled 17 volunteers who were clinically diagnosed with diabetes mellitus. The sample was assigned to one of two groups: the control group (CG, n = 7), who received information on foot care and health, and the intervention group (IG, n = 10), who received the application of foot reflexology on specific areas of the feet, for 10 consecutive days. There was blinding of the evaluator and the therapist. Surface electromyography (EMG) was used to assess the electrical activity of the medial and lateral gastrocnemius muscles in maximum voluntary isometric contraction (MVIC) and isotonic contraction (IC); baropodometry and stabilometry were used to analyze unloading, plantar weight distribution, and body sway. Results: There was a statistically significant difference for the variables of maximum peak electrical activity of the left medial gastrocnemius (*p* = 0.03; effect size = 0.87 and power = 0.81) and left lateral gastrocnemius muscles (*p* = 0.04, effect size = 0.70 and power = 0.66) respectively, in the intragroup IC, and median frequency of the left medial gastrocnemius muscle in the intragroup MVIC (*p* = 0.03; effect size = 0.64 and power = 0.59), and in the variables intergroups of the total area on the right side (*p* = 0.04; effect size = 1.03 and power = 0.50) and forefoot area on the left side (*p* = 0.02; effect size = 0.51 and power = 0.16). Conclusions: We conclude that foot reflexology influenced some variables of the intergroup plantar distribution and intragroup EMG in the sample studied. There is a need for a placebo group, a larger sample and a follow-up to strengthen the findings of these experiments.

## 1. Introduction

Medically speaking, diabetes mellitus (DM) represents a series of metabolic conditions associated with hyperglycemia; it is caused by partial or total insulin insufficiency [1]. Exposure to chronic hyperglycemia can result in microvascular complications in the retina, kidney, or peripheral nerves [1]. According to the International Diabetes Federation (IDF), 463 million people worldwide have diabetes, or 1 in every 11 adults aged 20–79 years [2]. This estimate is projected to reach 642 million by 2040, and studies suggest that the largest increases may happen in economically disadvantaged regions [2,3]. In 2017, DM was responsible for 4 million deaths globally, with an estimated expenditure of USD 727 billion, and it is among the top 10 causes of death in adults [2,3].

Individuals with type 2 diabetes mellitus (T2DM) who experience decompensation are more susceptible to developing both acute and chronic complications of T2DM [2]. The main complication is diabetic neuropathy (DN), which causes paresthesia, pain, and loss of sensation [2]. Paresthesia, pain, and loss of sensation may lead to several foot lesions and severe infections that increase the likelihood of reduced quality of life, amputations, and even early death [2].

Cutaneous feedback is important in regulating and modifying muscle activation patterns during gait [4]. Peak plantar pressures and patterns of muscle activations are altered when sensory feedback is reduced [4].

When analyzing peak plantar pressure and peak global and local shear (particularly in the hallux and center of the forefoot), in the ND group, the values were higher compared to non-neuropathic and healthy subjects [5].

DN causes delayed electromyographic activation and a longer period of electromyographic recording of the gastrocnemius and quadriceps muscles in relation to healthy individuals who will compromise postural control, as well as being an important factor contributing to the risk of falling [6,7].

Studies indicate that T2DM may lead to changes in postural balance; in the electrical activity of the muscles, resulting in muscle weakness and atrophy, bone deformities, and deviations in gait pattern. Such changes lead to a 5-fold higher risk of falls, reduced mobility, increased institutionalization rates, and mortality [5,6,7,8]. Changes in balance changes and muscle electrical activity can be assessed by baropodometry and surface electromyography, respectively, and predict diabetes-related outcomes [9,10].

Pharmacological agents and lifestyle changes are often essential in the prevention of comorbidities and management of patients with DM [11]. However, due to the significant side-effects of pharmacological agents, non-drug approaches are an attractive alternative [12]. Therefore, integrative and complementary practices have been used as options for health promotion and prevention, in addition to the rehabilitation of individuals with DM [11,12,13].

Integrative and complementary practices have currently become a reality in the public health care network [14]. From the First International Conference on Primary Health Care (Alma Ata, Russia, 1978) the World Health Organization (WHO) created the Traditional Medicine Program [14]. Within this approach, foot reflexology (FR) is a therapeutic and integrative method, which acts as a simple and easily applicable alternative; however, there remains little scientific evidence on the subject [11,12,13].

Foot reflexology is based on reflex zones, having, as a principle, the general balance of the body and an adequate energetic circulation of the organs and viscera [15]. Foot reflexology is performed by gentle pressure on specific parts of the feet that trigger functional responses in the body [15,16].

The application of manual therapy techniques, such as massage, joint mobilization and myofascial release, are capable of evoking a cascade of neurophysiological responses from the central and peripheral nervous systems, biomechanics and psychological responses arising from somatosensory stimuli [17,18]. These stimuli modulate pain responses, affective and somatoperceptual responses observed in clinical outcomes [17,18].

Meta-analytic studies and systematic reviews report that foot reflexology is a promising method for improving sleep, fatigue, pain, mood, nausea, and quality of life in several populations [19]. When foot reflex massage is applied to patients with diabetic neuropathy, it is able to reduce their complaints [20] modified the level of hemoglobins and increased peripheral vascular circulation [21]. improved general fatigue, heart rate and mood [22].

However, these studies show low quality of scientific evidence, low methodological quality and few randomized clinical trials. To date, there are no randomized clinical trial studies that analyzed the pattern of muscle activation and parameters of plantar pressures in the population with type 2 diabetic neuropathy that applied foot reflexology.

Thus, the aim of the present study was to verify the effect of foot reflexology on muscle electrical activity in the lateral and medial gastrocnemius muscles, distribution, plantar pressure, and body sway in patients with T2DM. We hypothesize that foot reflexology has a significant effect on muscle electrical activity, pressure, distribution, and body sway in patients with DM.

## 2. Materials and Methods

### 2.1. Study Design

This pilot randomized controlled clinical trial recruited participants between September 2019 and February 2020. All volunteers were informed about the objectives of the study, the importance of the activities developed, and the possible outcomes. Additionally, they voluntarily signed an informed consent form, according to the determinations of Resolution 466/12. This study was approved by the Research Ethics Committee of the Faculdade de Ciências Médicas (n°. 659.819) and was conducted in accordance with the Declaration of Helsinki [23]. This study obtained the clinical trial under the number RBR-7wjrkp9. This study followed the guidelines of the randomized clinical trial studies for the pilot study.

### 2.2. Sample Recruitment

The sample consisted of volunteers with a clinical diagnosis of T2DM recruited by telephone contact through a list of patients from the Basic Health Units and the Physical Therapy Clinic of the Universidade Federal de Alfenas, in the city of Alfenas, Minas Gerais, Brazil.

### 2.3. Eligibility Criteria

Volunteers of both genders with T2DM, aged over 40 years, with 3 years or more of diabetes onset, a visual analog pain scale with a moderate level (i.e., from mark 4 on the scale) [24], and who agreed to participate in the study and informed consent were included in the study. Volunteers who had type 1 DM or who had not been diagnosed with DM, who had constant use of muscle relaxant, who used a walking device, and who had lower extremity orthopedic and dermatologic problems, upper motor neuron injury, foot ulcers, deep vein thrombosis, or phlebitis, as well as those who declined to participate or declined to sign the informed consent were excluded.

### 2.4. Sample Randomization

Randomization was performed by a blinded researcher (Researcher 1). The names of the volunteers were numbered and placed in a brown, sealed envelope. Researcher 2 opened the envelope and performed the draw for which he/she allocated participants to one of two groups: the control group (CG) did not perform the intervention, and the intervention group (IG) performed the foot reflexology massage. The researcher who evaluated and the researcher who performed the intervention were blinded.

### 2.5. Study Location

The study was carried out in the physiotherapy clinic of Dr. Ana Cláudia Bonome Salate in the Human Performance Research Laboratory of the Universidade Federal de Alfenas—UNIFAL/MG, Santa Clara campus.

### 2.6. Sample Calculation

The sample calculation was performed with the program G Power, version 3.2, adopting an alpha value 0.05 and a power of 0.95. A pilot study was carried out with five individuals in each post-evaluation group and was determined by the variable total percentage (%) of weight on the right and left feet. In GI, it was 49 ± 2.12 on the left foot and 51 ± 5.06 on right foot. In CG, it was 55.20 ± 5.06 on the left foot and 44.8 ± 5.06 on the right foot. The effect size was 0.86. The total sample size was 10 individuals per group.

### 2.7. Sample Characterization

The sample was characterized by means of an evaluation form standardized by the researchers themselves containing the following clinical and anthropometric data: age, gender, weight, height, body mass index (BMI), time of DM diagnosis, shoe size, distal diabetic polyneuropathy [25], and foot pain [24].

### 2.8. Instruments Used for Evaluation

#### 2.8.1. Electromyography evaluation

##### Muscle Electrical Activity Data Collection

The evaluation of muscle electrical activity was performed by surface electromyography (EMG System do Brasil^®^, EMG-800C, São José dos Campus, Brazil). EMG contains an analog/digital conversion board with a 16-bit resolution, EMG amplifier with total amplification gain of 2000 times, a band-pass filter ranging from 20 to 500 Hz, eight unipolar active surface electrodes with gain pre-amplification of 20 times, a shielded cable and pressure clip at the end, common mode rejection >100 dB, and software for signal collection and analysis with a sampling frequency of 2000 Hz per channel. Using the Windows platform, common mode rejection ≥100 dB, preamplifier gain (cables) = gain 20 (with differential amplifier), gain of each channel = gain 100 times (configurable), system impedance = impedance 109 Ω, noise ratio = signal noise rate <3 μV RMS, hardware filters in the equipment = FPA (high pass) with a cut-off frequency of 20 Hz and FPB (low pass) with a cut-off frequency of 500 Hz were realized by a two-pole Butterworth-type analog filter.

The muscles evaluated were the lateral gastrocnemius (LG) and medial gastrocnemius (MG), bilaterally and simultaneously, during 5 sec of maximum voluntary isometric contraction (MVIC) and isotonic contraction (IC). Three repetitions were performed, with an interval of 1 min between each repetition. The data were analyzed by means of the root mean square (RMS) for MVIC, maximum peak for IC, and median frequency for MVIC and IC.

To place the electrodes, the volunteers were positioned in the prone position, with their knees extended and their feet out of the evaluation table. After performing skin asepsis with 70% alcohol and shaving, two 10 mm unipolar active electrodes (Meditrace^®^, Kendall, Macapá, Brazil) were adhered in the direction of the MG muscle fiber (longitudinally) in the most prominent muscle mass; the distance between the electrodes was 20 mm. The passive electrode was adhered to the right lateral malleolus. The clinical test of plantar flexion against a manual resistance performed by the evaluator was performed in order to verify the most prominent location of the muscle mass. To place the electrode on the LG muscle, the volunteers were positioned in the same way as for the analysis of the MG muscle. The electrodes were adhered between 1/3 of the line from the head of the fibula to the heel, in the direction of the muscle fiber (longitudinally). The collection of MG and LG muscle were performed simultaneously. All criteria of previous recommendations for the collection with surface electromyography were met [26].

The collection was performed with the volunteers in an orthostatic posture with bilateral barefoot support, with feet kept 15 cm apart, with knees and hips extended, and with the spine erect. The first three collections were performed while keeping the muscle at rest. Afterward, the volunteers were asked to perform the plantar flexion movement as a maximum contraction isometrically, and then isotonically.

##### EMG Data Analysis

Data were analyzed using MATLAB^®^ (verson 2017, the MathWorks Inc, Natick, MA, USA) software programming language. The signal was passed through a 20–500 Hz filter. Of the five seconds collected, the first and last seconds were excluded, leaving 3 s (corresponding to 3000 milliseconds of the sampling rate); this occurred for each collection.

For maximum voluntary isometric contraction, the amplitude of the signal was used in the time domain, the value given was in root mean square (RMS) and in the frequency domain, the value given was in the median frequency. The RMS value was obtained by calculating the average of the three collections; among the three collections, the highest value was selected, and the average of the three values was divided by the highest value of the collection and multiplied by 100%. The median frequency value was obtained by calculating the average of the three collections.

In the isotonic contraction, the amplitude of the signal was used in the time and frequency domains. In the time domain, the maximum peak was used and the median frequency was used for frequency. The maximum peak value was obtained through the average of the three collections; among the three collections, the highest value was selected, and the average of the three values was divided by the highest value of the collection and multiplied by 100%. To obtain the median frequency value, the average of the three collections was calculated [26].

#### 2.8.2. Distribution, Plantar Pressure, and Body Oscillation Data Collection

The evaluation of body oscillation behavior, unloading, and weight distribution in the right and left feet were performed using baropodometry and stabilometry electronic platform (Sensor Medica^®^, Guidonia Montecelio, Itália) connected to a computer by a USB cable with freeStep software. The platform has resistive sensors coated in 24-karat gold and conductive rubber; configurations were 40 cm/40 cm up to 300 cm/50 cm; USB was 2.0; sampling frequency was up to 400 Hz in real time; power supply was 15 Vdc; current consumption was 50–450 (mA); XY resolution was 2.5 dpi; Z resolution was 8 bits; dimensions were 440 mm/620 mm up to 3040 mm/740 mm; thickness was 8 mm; weight was 3.1–30 kg; it was of the matrix scanning type; automatic calibration was 10 bits; the suitable working temperature range was 0–55 °C; maximum pressure was 150 N/m^2^; and the sensor lifetime was 1,000,000 cycles.

For data collection, the platform was positioned 1 m from the wall, and a ruler with an angle of 15° was used to position the feet. The volunteers remained in the orthostatic posture, with bipodal support, barefoot, without dental occlusion and with ear occlusion, lower limbs kept extended, trunk erect, arms relaxed along the body, and fixed gaze, taking as reference a point on the wall at eye level [27,28]. The volunteers were first familiarized with the platform area, after which the distribution and plantar pressure were collected for 20 s, with the eyes open, in the aforementioned position. Afterward, the body sway was analyzed for 20 s with eyes open and closed, in the position mentioned above, with bipodal support. 

The variables analyzed in the distribution and plantar pressure were total surface area in cm^3^ on the right and left sides; surface area of the forefoot and hindfoot on the right and left sides; total weight load in % on the right and left sides; and weight load on the forefoot and hindfoot on the right and left sides. The body sway variable was bipodal total unloading in % with eyes open and closed on the right and left sides; weight unloading was on the forefoot on the right and left sides.

### 2.9. Procedures

The intervention group performed manual massage through foot reflexology by a researcher who was trained by a professional with more than 15 years of experience. She applied foot reflexology by performing deep pressure on reflex points throughout the foot for 15 min on each foot, totaling 30 min of intervention, which was performed for 2 consecutive weeks, excluding Saturday and Sunday.

For the application of the intervention, the volunteers were positioned in dorsal decubitus, with semi-flexed knees being supported by the “half-moon” pillow. The asepsis of the feet with 70% alcohol was previously performed, after which the blinded researcher performed the following steps: (1) stimulation of the solar plexus; (2) loosening the scapular girdle; (3) loosening the pelvic girdle; (4) stretching the tendon; (5) releasing the pelvic girdle; (6) pulling the neck musculature; (7) stretching the neck musculature; (8) releasing the tension of the head and neck; (9) relaxing the head; (10) relaxing the lung; (11) relaxing the diaphragm; (12) releasing the ribs; (13) relaxing the ribs; (14) releasing the vertebrae of the spine; and (15) the hugging maneuver (Figure 1) [29]. Mineral oil (Johnson’s^®^ Baby) was used for foot reflexology application, and the technique was performed in a quiet and comfortable environment, with minimal external interference. 

### 2.10. Statistical Analysis

The descriptive analysis of data was used to characterize the sample by mean and standard deviation for continuous variables and percentages for categorical variables. The Shapiro–Wilk test was applied to determine normality of the data, and independent *t*-tests were applied for the variables age, weight, height, BMI, shoes, time of diabetes, visual analog scale, EMG, distribution and plantar pressure, and body sway, whereas the Chi-square test was applied for the variables gender, DN, culturing sensitivity, neuropathic symptoms, and neuropathic impairment. Paired *t*-tests were used for the variables EMG, plantar distribution and pressure, and body sway. All analyses were performed by the SPSS statistical software, version 20.0, IBM, New York, NY, USA. 

## 3. Results

Volunteers were recruited via telephone call. A total of 70 patients were contacted, of which 49 were excluded due to the following reasons: no transportation (n = 12), declined to participate (n = 19), did not answer the call (n = 15), and other reasons (n = 3). Thus, 21 patients were selected for evaluation; of these, 4 were excluded for failing to meet the inclusion criteria. Therefore, the final study sample had a total of 17 volunteers who were randomized into one of two groups: the control group (CG, n = 7) or the intervention group (IG, n = 10). There was no sample loss in either group. The figure below shows the sample arrangement (Figure 1).

Table 1 shows the demographic and clinical data of the CG and IG regarding the variables. A statistically significant difference is seen between the groups with respect to plantar skin sensitivity of the left foot.

Table 2 shows the mean, standard deviation, confidence interval (CI) and intra- and inter-group p-values of the surface electromyography (EMG) variables in the maximum voluntary isometric contraction (MVIC). In the MVIC, the data were analyzed by the root mean square (RMS) and by the median frequency (FREQ) of the CG and IG.

Table 2 shows a statistically significant intragroup difference in the median frequency of the left medial gastrocnemius (*p* = 0.03), and intergroup, there was no statistically significant difference in the variables analyzed (*p* > 0.05). The mean difference for the median frequency variable for the left GM was 144.65 post-intervention, and the confidence interval was between 104.32 and 184.97. Note that the confidence interval values did not pass through 0, indicating an effect produced by the technique in relation to pre-intervention and post-intervention. The effect size was 0.64, indicating a medium effect size. The power was 0.59, indicating low power.

Table 3 shows the mean, standard deviation, confidence interval and intra- and inter-group *p*-values of the surface electromyography (EMG) variables in isotonic contraction (IC). In the CI, the data were analyzed by the maximum peak (MAX) and median frequency (FREQ) for the GC and IG.

In Table 3, it is noted that there was a statistically significant difference within the intragroup in the variables maximum peak of the left lateral gastrocnemius and left medial gastrocnemius (*p* = 0.04 and *p* = 0.03, respectively), and there was no intergroup difference in the variables analyzed (*p* > 0.05). 

The mean of the variable maximum peak of the left lateral gastrocnemius was 82.55 after the intervention, the confidence interval was between 72.26 and 92.84. The mean of the maximum peak variable of the left medial gastrocnemius was 83.62 post-intervention, and the confidence interval was between 75.53 and 91.72. Note that the confidence interval did not pass the value 0, indicating an effect of the technique in relation to pre-intervention and post-intervention. The effect size for a left lateral gastrocnemius maximum peak variable was 0.70, indicating a mean effect size. The power was 0.66, indicating low power. The effect size for a maximal left medial gastrocnemius peak variable was 0.87, indicating a high effect size. The power was 0.81, indicating high power.

Table 4 presents the mean, standard deviation, and 95% confidence interval of the distribution and plantar pressure variables, for CG and GI. A statistically significant difference is seen for the total surface area of the left foot and surface area of the left forefoot intergroup variables (*p* > 0.05). We then determined the differences in the means between groups for total surface area of the left foot variable (17.24) with confidence interval (0.44 to 34.04) and surface area of the left forefoot variable (7.65) with confidence interval (−7.68 to 2.28). Note that the confidence interval values did not pass through 0, indicating an effect produced by the technique for the variable total surface area of the left. The effect size for the total surface area of the left foot variable was 1.03, indicating a high effect size. The power was 0.50, indicating low power. The effect size for the surface area of the left forefoot variable was 0.51, indicating a low effect size. The power was 0.16, indicating low power.

Table 5 shows the mean, standard deviation, confidence interval, and intra- and intergroup *p*-value of the body sway variables for CG and IG. No statistically significant difference was found in the variables analyzed (*p* > 0.05).

## 4. Discussion

The main finding of the present study is that foot reflexology generated effects on the variables of maximum peak electrical activity of the left medial gastrocnemius and left lateral gastrocnemius muscles, respectively, in the intragroup isotonic contraction, and median frequency of the left medial gastrocnemius muscle in the intragroup maximum voluntary isometric contraction, and on the intergroup variables of total surface area of the right side and forefoot surface area of the left side of patients with T2DM. This finding could be explained by the touch produced by foot reflexology after the intervention [16]. Massage activates receptors for pressure, temperature, and nociception, as well as increasing local blood flow and muscle relaxation in the cutaneous area of the feet [13,30].

Studies indicate that massage is capable of producing pain modulation, reducing muscle spasms and joint stiffness [18,30]. The tactile information evoked by massage stimulates the fast and large-caliber nerve fibers (Aβ and Aδ), inhibiting the slow, small-caliber fibers, consequently reducing the perception of pain [18], as well as increasing the release of neurotransmitters and thus, playing an important role in modulation [30]. Massage-induced mechanical pressure generated changes in muscle–tendon compliance by mobilizing and stretching connective cells, improving joint and muscle stiffness [30].

The act of touching the surface of the skin evokes deactivation of the system related to the stress and threat response, that is, generating affective effects. Massage can also improve body perceptions through the reorganization of mental representations of the body [18].

Results obtained in the total surface area show an increase in the contact area on the right side of the foot and a reduction on the left side in IG, indicating improvement in the distribution of plantar contact surface when comparing the right side with the left side. In the forefoot surface area, there is a reduction in the values in the GI, translating a better distribution of the plantar contact surface. These findings could be explained by the reduction in pain symptoms, which could improve the efficiency of the muscular, articular, and vestibular proprioceptors, especially in the occlusion of the eyes and hearing [31,32,33].

The results found in the study by Silva et al. [13] show no effects of foot reflexology on the variable plantar surface area in patients with diabetes, contradicting the findings of the present study. There is no evidence to support why the contradictions were seen.

The findings of Megda et al. [33] agree with the findings of the present study, who verified the immediate effect of reflexology on the variable mass division in the plantar area with eyes closed in patients with diabetic neuropathy. Yümin et al. [32] obtained positive responses in relation to mobility, balance and functional range after the application of foot massage in patients with diabetic neuropathy, corroborating the findings of the present study.

The degree of impairment of DN, long-term inadequate glycemic control, age, gender and the presence of other complications or comorbidities are also factors that may impact balance [34]. Analyzing the population of the present study, we noticed that the mean age was older than 60 years, and the majority had DN, altered sensitivity, especially on the left side, and with a time of more than 10 years with a diagnosis of diabetes.

The results found in the muscle electrical activity variables denote a lower muscle activation in isotonic and isometric contractions. This result can be justified due to the fact that the muscular components and the neural components of the motor units in patients with T2DM are altered both in their structure and functionality; these alterations may be associated with insulin resistance [35].

The alterations found in the neural components in both structure and function are present in the central sensitization syndrome, which are observed in neuropathies [36]. Permanent central sensitization is considered an expression of neuronal plasticity in primary sensory neurons and spinal dorsal horn neurons [37].

Rusu et al. [38] described that in patients with DM, the number of muscle fibers does not change, but their physiological, mechanical, and biochemical properties may change, showing that the metabolic complications can cause electromyographic changes, even in the early stages of the disease.

The present study has the following limitations: the need for a placebo group, as the ability to explore the findings and draw conclusions was limited; and the sample size was smaller than necessary to obtain statistical significance of the results, although the results indicate clear trends for the intervention group. Future work to increase the sample size to obtain statistically significant results and follow-up from participants after longer periods of time would strengthen the findings of these experiments.

The clinical implications in the present study are being pointed out as a low-cost technique that is easily applied by professionals in the public health network.

## 5. Conclusions

We conclude that foot reflexology influenced some variables of intergroup plantar distribution and intragroup EMG in the sample studied. There is a need for a placebo group, a larger sample and a follow-up to strengthen the findings of these experiments.

## Figures and Tables

**Figure 1 ijerph-19-14547-f001:**
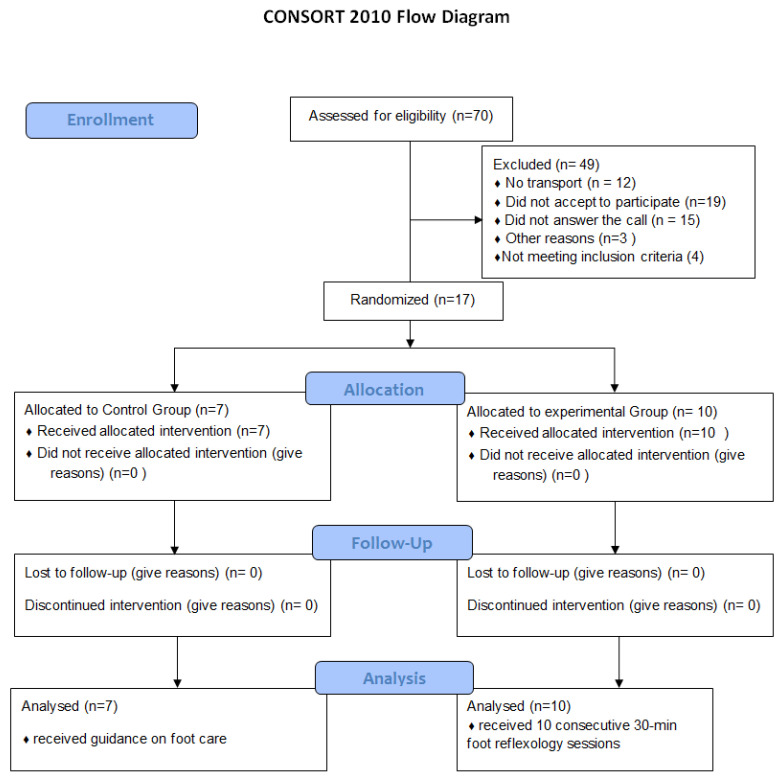
Flow diagram based on CONSORT 2010.

**Table 1 ijerph-19-14547-t001:** Sociodemographic and clinical characteristics of study participants.

Characteristics (Average (SD))	CG (n = 7)		IG (n = 10)	*p* Value
Age (years)	58 ± 6.81		65 ± 12.66	0.24 ^a^
Height (m)	1.62 ± 0.06		1.58 ± 0.13	0.51 ^a^
Body Mass (kg)	84.38 ± 16.80		72.28 ± 10.43	00.16 ^a^
BMI cm/kg	32.04 ± 7.09		28.91 ± 4.34	0.37 ^a^
Footwear number	38.33 ± 0.81		38 ± 2.12	0.79 ^a^
Diagnosis time (years)	14 ± 10.23		15 ± 7.54	0.72 ^a^
Analogic Visual Scale	8.00 ± 1.91		8.00 ± 1.41	0.68 ^a^
Sex (%)	F	5–71.4	6–60.00	0.62 ^b^
	M	2–28.6	4–40.00	
Diabetic Neuropathy (%)	S	5–71.4	9–90.00	0.32 ^b^
	N	2–28.6	1–10.00	
Skin sensitivity plant right (%)	P	4–57.1	7–70.00	0.58 ^b^
	A	3–42.9	3–30.00	
Skin sensitivity plant left (%)	P	2–28.6	8–80.00	**0.03 ^b^***
	A	5–71.4	2–20.00	
Neuropathic symptom score (%)	L	1–14.28	1–10.00	0.78 ^b^
	M	3–42.8	6–60.00	
	G	3–42.8	3–30.00	
Neuropathic impairment score (%)	L	4–57.10	3–80	0.49 ^b^
	M	1–14.3	2–20	
	G	0	0	
	NC	2–28.6	5–50	

^a^*t* test, ^b^ chi-square test; Legend: BMI: body mass index; CG: control group; IG: intervention group; F: female; M: male; S: yes; N: no; P: preserved; A: altered; L: light; M: moderate; G: severe; NC: did not classify. * *p* value < 0.05.

**Table 2 ijerph-19-14547-t002:** Mean value, standard deviation and confidence interval in the comparison of GC and IG for electromyography variables in maximum voluntary isometric contraction.

	Group				*p* Value Difference between Times ^a^		*p* Value Difference between Groups ^b^	
	Pre Intervention		Post Intervention					
Variable Gastrocnemius Muscle (%)	CG (n = 7) IC 95%	IG (n = 10) IC 95%	CG (n = 7) IC 95%	IG (n = 10) IC 95%	CG	IG	CG/IG Pre Intervention	CG/IG Post Intervention
Root mean square for the right lateral gastrocnemius	88.98 ± 8.36 81.26–96.73	83.29 ± 14.12 73.19–93.40	92.87 ± 3.17 89.93–95.05	85.96 ± 14.24 75.76–96.15	0.31	0.39	0.35	0.16
Median frequency of the right lateral	179.40 ± 65.37 118.93–239.86	171.64 ± 63.39 126.29–216.99	138.61 ± 44.65 97.31–179.92	142.24 ± 54.38 103.34–181.15	0.13	0.85	0.81	0.88
Root mean square for the left lateral	91.97 ± 3.27 88.94–95.00	87.13 ± 12.20 78.40–95.86	91.77 ± 4.14 87.94–95.60	88.93 ± 12.91 79.69–98.16	0.94	0.51	0.32	0.58
Median frequency of the left lateral	184.47 ± 73.34 116.63–252.30	172.16 ± 43.66 140.92–203.39	143.99 ± 40.28 106.73–181.25	156.50 ± 52.57 118.89–194.11	0.14	0.20	0.67	0.60
Root mean square for the right medial	88.78 ± 7.78 81.58–95.98	90.42 ± 7.95 84.73–96.11	90.27 ± 5.45 85.22–95.32	89.21 ± 9.63 82.31–96.10	0.66	0.51	0.67	0.79
Median frequency of the right medial	178.93 ± 66.07 117.83–240.04	156.50 ± 67.85 107.96–205.04	146.52 ± 38.95 110.49–182.55	147.91 ± 52.24 110.53–185.28	0.23	0.72	0.50	0.95
Root mean square for the left medial	89.34 ± 8.01 81.92–96.75	91.45 ± 7.02 86.43–96.48	93.04 ± 4.49 88.87–97.20	92.21 ± 5.54 88.24–96.18	0.32	0.77	0.57	0.75
Median frequency of the left medial	186.37 ± 51.71 138.55–234.20	182.18 ± 60.28 139.05–225.31	159.68 ± 41.77 121.04–198.32	144.65 ± 56.37 104.32–184.98	0.32	**0.03** *	0.88	0.55

**Legend:** ^a^ paired *t* test. ^b^ independent *t* test; IG: intervention group; CG: control group; 95% CI: 95% confidence interval, * *p* value < 0.05.

**Table 3 ijerph-19-14547-t003:** Mean values, standard deviation and confidence interval in the comparison of GC and IG for electromyography variables in isotonic contraction.

	Group				*p* Value Difference between Times ^a^		*p* Value Difference between Groups ^b^	
	Pre Intervention		Post Intervention					
Variable Gastrocnemius Muscle (%)	CG (n = 7) IC 95%	IG (n = 10) IC 95%	CG (n = 7) IC 95%	IG (n = 10) IC 95%	CG	IG	CG/IG Pre Intervention	CG/IG Post Intervention
Maximum peak for the right lateral	85.79 ± 79 78.95–92.63	88.36 ± 5.38 84.51–92.22	82.70 ± 13.74 69.99–95.41	84.13 ± 13.48 74.48–93.77	0.59	0.45	0.41	0.83
Median frequency of the right lateral	157.31 ± 73.39 89.43–225.19	162.10 ± 72.33 110.36–213.85	140.62 ± 65.36 80.17–201.06	127.95 ± 37.29 101.28–154.63	0.64	0.09	0.89	0.61
Maximum peak for the left lateral	93.89 ± 3.19 90.93–96.84	91.77 ± 3.46 89.29–94.24	83.31 ± 16.29 67.93–98.70	82.55 ± 14.38 72.26–92.84	0.18	**0.04 ***	0.22	0.92
Median frequency of the left lateral	170.19 ± 73.51 102.20–238.18	146.96 ± 36.99 120.50–173.43	137.36 ± 50.23 90.90–183.82	150.22 ± 60.76 106.75–193.69	0.46	0.84	0.40	0.65
Maximum peak for the right medial	89.36 ± 4.24 85.43–93.28	91.29 ± 6.72 86.48–96.11	81.32 ± 16.29 66.25–96.39	85.58 ± 11.75 77.17–93.99	0.30	0.19	0.51	0.53
Median Frequency of the right medial	153.68 ± 75.85 83.53–223.84	150.02 ± 70.69 99.45–200.60	121.23 ± 20.58 102.19–140.26	122.13 ± 42.83 91.49–152.77	0.36	0.17	0.92	0.96
Maximum peak for the left medial	87.36 ± 6.55 81.29–93.42	92.29 ± 4.36 89.17–95.41	82.11 ± 13.77 69.37–94.85	83.62 ± 11.31 75.53–91.72	0.45	**0.03 ***	0.08	0.80
Median frequency of the left medial	166.94 ± 70.15 102.05–231.82	161.61 ± 64.44 115.51–207.72	134.34 ± 30.41 106.21–162.47	127.47 ± 45.71 94.76–160.17	0.42	0.11	0.87	0.73

**Legend:** ^a^ paired *t* test. ^b^ independent *t* test; IG: intervention group; CG: control group; IC 95%: 95% interval confidence. * *p* value < 0.05.

**Table 4 ijerph-19-14547-t004:** Mean values, standard deviation and confidence interval in the comparison of GC and IG for distribution variables and plantar pressure.

	Group				*p* Value Difference between Times ^a^		*p* Value Difference between Groups ^b^	
	Pre Intervention		Post Intervention					
Variable	CG (n = 7) CI 95%	IG (n = 10) CI 95%	CC (n = 7) CI 95%	IG (n = 10) CG 95%	CG	IG	CG/IG Pre Intervention	CG/IG Post Intervention
Total surface area of the left foot (cm^3^)	122.28 ± 14.25 109.10–135.47	111.00 ± 17.78 98.27–123.72	124.42 ± 25.49 100.84–148.00	107.90 ± 19.93 93.64–122.15	0.73	0.42	0.18	0.15
Total surface area of the right foot (cm^3^)	118.14 ± 18.49 101.03–135.24	103.10 ± 16.15 91.54–114.65	119.14 ± 19.47 101.13–137.15	101.90 ± 13.17 92.47–111.32	0.86	0.67	0.09	**0.04 ***
Full weight bearing on the left foot (%)	51.14 ± 8.23 43.52–58.75	51.00 ± 6.58 46.29–55.70	52.57 ± 6.21 46.82–58.31	51.70 ± 4.69 48.34–55.05	0.70	0.72	0.96	0.74
Full weight bearing on the right foot (%)	48.85 ± 8.23 41.24–56.47	49.00 ± 6.58 44.29–53.70	47.42 ± 6.21 41.68–53.17	48.30 ± 4.69 44.94–51.65	0.70	0.72	0.29	0.30
Surface area of the left forefoot (cm^3^)	66.42 ± 9.03 58.07–74.78	60.40 ± 12.35 51.55–69.24	66.85 ± 15.52 52.49–81.21	59.20 ± 13.95 49.22–69.17	0.91	0.70	0.12	**0.02 ***
Surface area of the right forefoot (cm^3^)	63.28 ± 13.42 50.86–75.70	54.30 ± 9.42 47.55–61.04	64.85 ± 11.58 54.14–75.56	53.10 ± 8.45 47.05–59.14	0.75	0.57	0.65	0.53
Weight bearing on the left forefoot (%)	23.85 ± 4.52 19.67–28.04	22.70 ± 5.49 18.76–26.61	25.28 ± 4.07 21.52–29.05	23.70 ± 5.69 19.62–27.77	0.39	0.65	0.40	0.43
Weight bearing on the right forefoot (%)	22.57 ± 5.56 17.42–27.71	20.20 ± 5.65 16.15–24.24	21.85 ± 3.67 18.46–25.25	20.00 ± 5.31 16.19–23.80	0.80	0.90	0.14	0.07
Surface area of the left hindfoot (cm^3^)	56.00 ± 8.18 48.42–63.57	50.50 ± 6.45 45.88–55.11	57.42 ± 11.31 46.96–67.89	48.70 ± 7.33 43.45–53.94	0.62	0.68	0.13	0.17
Surface area of the right hindfoot (cm^3^)	54.71 ± 8.05 47.26–62.16	48.60 ± 7.63 43.13–54.06	54.28 ± 9.10 45.86–62.70	48.80 ± 6.90 43.85–53.74	0.85	0.90	0.74	0.75
Weight bearing on the left hindfoot (%)	27.28 ± 8.49 19.42–35.14	28.30 ± 3.94 25.47–31.12	27.28 ± 5.37 22.31–32.25	28.00 ± 3.74 25.32–30.67	1	0.78	0.28	0.25
Weight bearing on the right hindfoot (%)	26.28 ± 4.23 22.37–30.19	28.80 ± 4.84 25.33–32.26	25.57 ± 3.35 22.46–28.46	28.30 ± 5.31 24.49–32.10	0.71	0.81	0,28	0,25

**Legend**: ^a^ paired *t* test. ^b^ independent *t* test; IG: intervention group; CG: control group; 95% CI: 95% confidence interval, * *p* value < 0.05.

**Table 5 ijerph-19-14547-t005:** Mean values, standard deviation and confidence interval in the comparison of GC and IG for the body sway variables.

	Group				*p* Value Difference between Times ^a^		*p* Value Difference between Groups ^b^	
	Pre Intervention		Post Intervention					
Variable (%)	CG (n = 7) 95% CI	IG (n = 10) 95% CI	CG (n = 7) 95% CI	IG (n = 10) 95% CI	CG	IG	CG/IG Pre Intervention	CG/IG Post Intervention
Bipedal open eye full weight bearing left	51.14 ± 6.44 45.18–57.09	53.10 ± 7.93 47.42–58.77	45.71 ± 20.63 26.62–64.80	53.30 ± 6.70 48.50–58.09	0.45	0.91	0.59	0.29
Bipedal open eye full weight bearing right	48.85 ± 6.44 42.90–54.81	46.90 ± 7.93 41.22–52.57	54.28 ± 20.63 35.19–73.37	46.70 ± 6.70 41.90–51.49	0.45	0.91	0.59	0.29
Bipedal open eye forefoot weight-bearing left	41.00 ± 8.75 32.90–49.09	39.50 ± 9.10 32.98–46.01	36.57 ± 16.94 20.89–52.04	41.50 ± 10.25 34.16–48.83	0.46	0.53	0.73	0.46
Bipedal open eye forefoot weight-bearing right	36.57 ± 6.82 30.25–42.88	35.80 ± 8.94 29.40–42.19	29.00 ± 14.88 15.23–42.76	36.60 ± 10.49 29.09–44.10	0.25	0.73	0.85	0.23
Bipedal eye open hindfoot weight bearing left	59.00 ± 8.75 50.90–67.09	60.50 ± 9.10 53.98–67.01	49.14 ± 22.28 28.52–69.75	58.50 ± 10.25 51.16–65.83	0.32	0.53	0.73	0.25
Bipedal open eye weight bearing hindfoot right	63.42 ± 6.82 57.11–69.74	64.20 ± 8.94 57.80–70.59	71.00 ± 14.88 57.23–84.76	63.40 ± 10.49 55.89–70.90	0.25	0.73	0.85	0.23
Bipedal closed eye full weight bearing left	51.14 ± 5.55 46.00–56.27	51.70 ± 7.52 46.31–57.08	47.14 ± 21.80 26.97–67.30	53.20 ± 6.37 48.64–57.75	0.59	0.08	0.87	0.41
Bipedal closed eye full weight bearing right	48.85 ± 5.55 43.72–53.99	48.30 ± 7.52 42.91–53.68	52.85 ± 21.80 32.69–73.02	48.70 ± 4.76 45.29–52.10	0.59	0.86	0.87	0.56
Bipedal closed eye forefoot weight-bearing left	43.57 ± 9.43 34.84–52.29	41.70 ± 10.57 34.13–49.26	38.00 ± 19.54 19.94–56.07	44.80 ± 10.50 37.28–52.31	0.35	0.24	0.71	0.36
Bipedal closed eye forefoot weight-bearing right	40.14 ± 4.22 36.23–44.04	38.30 ± 11.49 30.07–46.52	33.71 ± 15.81 19.08–48.34	39.20 ± 9.98 32.05–46.34	0.30	0.68	0.65	0.39
Bipedal closed eye weight bearing hindfoot left	56.42 ± 9.43 47.70–65.15	58.30 ± 10.57 50.73–65.86	47.71 ± 23.32 26.14–69.28	55.20 ± 10.50 47.68–62.71	0.39	0.24	0.71	0.38
Bipedal closed eye weight bearing hindfoot right	59.85 ± 4.22 55.95–63.76	61.70 ± 11.49 53.47–69.92	66.28 ± 15.81 51.65–80.91	60.80 ± 9.98 53.65–67.94	0.30	0.68	0.65	0.39

**Legend:** ^a^ paired *t* test. ^b^ independent *t* test; IG: intervention group; CG: control group; 95% CI: 95% confidence interval.

## Data Availability

Not applicable.

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
