# Peer review of "Effect of Foot Reflexology on Muscle Electrical Activity, Pressure, Plantar Distribution, and Body Sway in Patients with Type 2 Diabetes Mellitus: A Pilot Randomized Controlled Trial"

_ijerph, 2022, doi:10.3390/ijerph192114547_

Round 1

Reviewer 1 Report

I read your study with interest as early low cost interventions for diabetics could be important for lifelong health and well being.

I think that your study is interesting and relevant but the presentation and description is a little confusing in places.

I would suggest the following minor changes to improve your work

1) In the introduction could you explain the importance of interventions to change muscle activity and foot pressure in diabetics - this would explain the impact of your work better

2) Also in the introduction could you give some references of why foot pressure and muscle activation are important for diabetics - again this could help to better explain the importance of your work.

Some suggested references would be:

"The speed at which joint strength is produced is an important factor related to unsteadiness, with a slower generation of joint strength indicative of a higher risk of falling"

* Bento   PC Pereira   G Ugrinowitsch   C Rodacki   AL .  Peak torque and rate of torque development in elderly with and without fall history .  Clin Biomech (Bristol, Avon)   2010 ; 25 : 450 – 454   * LaRoche   DP Cremin   KA Greenleaf   B Croce   RV .  Rapid torque development in older female fallers and nonfallers: a comparison across lower-extremity muscles .  J Electromyogr Kinesiol   2010 ; 20 : 482 – 488   * Pijnappels   M Reeves   ND Maganaris   CN van Dieën   JH .  Tripping without falling; lower limb strength, a limitation for balance recovery and a target for training in the elderly .  J Electromyogr Kinesiol   2008 ; 18 : 188 – 196     "Patients with DPN are slower at generating strength at the ankle and knee than control participants" * Joseph C. Handsaker, Steven J. Brown, Frank L. Bowling, Glen Cooper, Constantinos N. Maganaris, Andrew J.M. Boulton, Neil D. Reeves; Contributory Factors to Unsteadiness During Walking Up and Down Stairs in Patients With Diabetic Peripheral Neuropathy. Diabetes Care 1 November 2014; 37 (11): 3047–3053.   "It is believed that the primary etiologic factors for DFU are repetitive stresses in the presence of peripheral neuropathy" M. Yavuz Plantar shear stress distributions in diabetic patients with and without neuropathy Clin. Biomech. Bristol Avon., 29 (2014), pp. 222-223   Adding these references with a better narrative about the importance of your work would help improve impact   3) In the tables in the results section - there are a lot of abreviations in the titles - I would prefer these to be written in full so readers can access these without reading the full text   4) Also other specific comments are shown below:    

Line 28 - Can you define FR please - also line 66 - personally I would write this in full instead

Line 143 - Can you give more detail about electrode placement? E.g. distal/proximal or 2/3rds along or something similar. Also could you give an indication of the size of the electrode used, e.g. diameter of electrode, also the position of the ground electrode (in line 126 you talk about bipolar electrodes - can you be clear on this for the reader). I would also suggest that you use a subheading for this section "EMG Data Collection" or similar

Line 150 - Matlab is a programming language not an EMG processing program. Did you write your own custom code to do the processing? If so can you describe the sampling rate of your data, the type of filtering/windowing that you used, whether it is single pass etc. Again I would suggest that you put a subheading of something like "EMG Data Analysis" for this section.

Line 152 - do you mean that the RMS and maximum amplitude for each of the three data collections were calculated. Then the average RMS and Maximum Amplitude were calculated from the three results? At the moment this is a little confusing - could you rephrase please

Line 183 - FR zones - could you be more specific - you need to mention figure 1 in the text please

Line 272 - Don't understand how this can be explained by the touch produced by FR? Do you mean that after the FR intervention you measured differences?

Line 267 - 275 - Can you be more specific - E.g. were the effects observed to be 20% higher or 10% lower - please be specific for the reader. Also is this for the control group or the diabetic group?

Line 281-283 - Why are these contradictions present? Do you have any insight - if not please say there is no evidence to support why contradictions were seen.

Line 297-302 - I would suggest that you rephrase your limitations section - at present it reads as an excuse rather than a scientific explanation. Every study has limitations. I would suggest writing something like:

"The present study has limitations. The sample size was smaller than needed to obtain statistical significance of the results although the results do indicate clear trends. Practically and ethically the sample size could not be increased due to the Covid-19 pandemic and test subjects belonging to an at risk group. Future work to increase the sample size to obtain statistically significant results and follow-up of participants after longer time periods would strengthen findings from these experiments."

Author Response

Reviewer's Suggestions 1

First I would like to thank you for the suggestions. The suggestions were very important for the improvement of the manuscript. Below I have described item by item in order of correction

1) In the introduction could you explain the importance of interventions to change muscle activity and foot pressure in diabetics - this would explain the impact of your work better

2) Also in the introduction could you give some references of why foot pressure and muscle activation are important for diabetics - again this could help to better explain the importance of your work.

Answer: this text has been inserted

Cutaneous feedback is important in regulating and modifying muscle activation patterns during gait [4]. Peak plantar pressures and patterns of muscle activations are altered when sensory feedback is reduced [4].

When analyzing peak plantar pressure and peak global and local shear (in particular in the hallux and center of the forefoot), in the ND group the values were higher compared to non-neuropathic and healthy subjects [5].

DN causes delayed electromyographic activation and a longer period of electromyographic recording of the gastrocnemius and quadriceps muscles in relation to healthy individuals who will compromise postural control, as well as being an important factor contributing to the risk of falling [6,7]

3) In the tables in the results section - there are a lot of abreviations in the titles - I would prefer these to be written in full so readers can access these without reading the full text  

The tables have been corrected in the text. All removed the abbreviations

4) Also other specific comments are shown below:    

Line 28 - Can you define FR please - also line 66 - personally I would write this in full instead

All abbreviations in the text referring to RF were written

Line 143 - Can you give more detail about electrode placement? E.g. distal/proximal or 2/3rds along or something similar. Also could you give an indication of the size of the electrode used, e.g. diameter of electrode, also the position of the ground electrode (in line 126 you talk about bipolar electrodes - can you be clear on this for the reader). I would also suggest that you use a subheading for this section "EMG Data Collection" or similar

To place the electrodes, the volunteers were positioned in the prone position, with their knees extended and their feet out of the evaluation table. After performing skin asepsis with 70% alcohol and shaving, two 10mm unipolar active electrodes (Meditrace®) were adhered in the direction of the MG muscle fiber (longitudinally) in the most prominent muscle mass, the distance between the electrodes was 20mm. The passive electrode was adhered to the right lateral malleolus. The clinical test of plantar flexion against a manual resistance performed by the evaluator was performed, in order to verify the most prominent location of the muscle mass. To place the electrode on the LG muscle, the volunteers were positioned in the same way as for the analysis of the MG muscle. The electrodes were adhered between 1/3 of the line from the head of the fibula to the heel, in the direction of the muscle fiber (longitudinally). The collection of MG and LG muscle were performed simultaneously.  All criteria of previous recommendations for the collection with surface electromyography were met [26].

Subtitle was used for this section

 2.8.1.1 Muscle electrical activity data collection

Line 150 - Matlab is a programming language not an EMG processing program. Did you write your own custom code to do the processing? If so can you describe the sampling rate of your data, the type of filtering/windowing that you used, whether it is single pass etc. Again I would suggest that you put a subheading of something like "EMG Data Analysis" for this section.

Line 152 - do you mean that the RMS and maximum amplitude for each of the three data collections were calculated. Then the average RMS and Maximum Amplitude were calculated from the three results? At the moment this is a little confusing - could you rephrase please

2.8.1.2 EMG data analysis

Data were analyzed using MATLAB® software programming language. The signal was passed through a 20-500 Hz filter. Of the five seconds collected, the first and last second were excluded, leaving 3 seconds (corresponding to 3000 milliseconds of the sampling rate), this occurred for each collection.           

In maximum voluntary isometric contraction, the amplitude of the signal was used in the time domain, the value given was in root mean square (RMS) and in the frequency domain, the value given was in median frequency. The RMS value was obtained by calculating the average of the three collections, among the three collections, the highest value was selected and the average of the three values ​​was divided by the highest value of the collection and multiplied by 100%. The median frequency value was obtained by calculating the average of the three collections.

In isotonic contraction used the amplitude of the signal in the time domain, the value was given at maximum peak in the frequency domain the value given was in the median frequency. The maximum peak value was obtained through the average of the three collections, among the three collections, the highest value was selected and the average of the three values ​​was divided by the highest value of the collection and multiplied by 100%. To obtain the median frequency value, the average of the three collections was calculated. [26].

Line 183 - FR zones - could you be more specific - you need to mention figure 1 in the text please

The term reflex zone was replaced by reflex point and inserted in figure 1

She applied foot reflexology by performing deep pressure on in reflex points (figure 1) throughout the foot for 15 min on each foot, totaling 30 min of intervention, which was performed for 2 consecutive weeks, excluding Saturday and Sunday.

Line 272 - Don't understand how this can be explained by the touch produced by FR? Do you mean that after the FR intervention you measured differences?

This finding could be explained by the touch produced by foot reflexology after the intervention

Line 267 - 275 - Can you be more specific - E.g. were the effects observed to be 20% higher or 10% lower - please be specific for the reader. Also is this for the control group or the diabetic group?

Results obtained in the total surface area show an increase in contact area on the right side of the foot and reduction on the left side in IG, indicating improvement in the distribution of plantar contact surface when comparing the right side with the left side. In the forefoot surface area, there is a reduction in the values in the GI, translating a better distribution of the plantar contact surface.

Line 281-283 - Why are these contradictions present? Do you have any insight - if not please say there is no evidence to support why contradictions were seen.

The results found in the study by Silva et.al. [13] show no effects of foot reflexology on the variable plantar surface area in patients with diabetes, contradicting the findings of the present study. There is no evidence to support why the contradictions were seen.

Line 297-302 - I would suggest that you rephrase your limitations section - at present it reads as an excuse rather than a scientific explanation. Every study has limitations. I would suggest writing something like:

"The present study has limitations. The sample size was smaller than needed to obtain statistical significance of the results although the results do indicate clear trends. Practically and ethically the sample size could not be increased due to the Covid-19 pandemic and test subjects belonging to an at risk group. Future work to increase the sample size to obtain statistically significant results and follow-up of participants after longer time periods would strengthen findings from these experiments."

The present study has the following limitations: the need for a placebo group, as the ability to explore the findings and draw conclusions was limited; the sample size was smaller than necessary to obtain statistical significance of the results, although the results indicate clear trends for the intervention group. Future work to increase the sample size to obtain statistically significant results and follow-up from participants after longer periods of time would strengthen the findings of these experiments.

Reviewer 2 Report

Dear Authors

Thanks a lot for the opportunity you have offered me to revise the fascinating manuscript " Effect of foot reflexology on muscle electrical activity, pressure,

plantar distribution, and body sway in patients with type 2 diabetes mellitus: a randomized clinical trial". I thank the authors for their effort in producing this exciting manuscript. It is perfectly aligned with my area of research and expertise; thus, I am confident to offer a valuable peer review.

As a significant strength, this manuscript discusses the Effect of foot reflexology on the electrical muscle activity of the lateral and medial gastrocnemius muscle and examines the distribution, plantar pressure, and body sway in patients with type 2 diabetes mellitus. This proposal is a novelty in the field and adds information to the existing evidence in the literature produced in the field.

As a major weakness, the manuscript sometimes lacks details and clarity concerning methodological steps that would help improve the understanding of the manuscript. Moreover, at a general level, the manuscript seems to be more of a pilot RCT than a large trial. Therefore, I have suggested some strategies to improve authors' reporting and increase the quality of their work (e.g., rationale/background and discussion of the manuscript).

Overall, my peer-review is a major revision: I suggest revising the manuscript to improve the pitfalls presented. The final goal is to improve the overall clarity of the message to help the reader understand this fundamental topic.

I look forward to reading the revised version of the manuscript.

Thanks again, and good luck with researching in this challenging time.

#TITLE:

* I suggest changing the title, adding “: a pilot randomized controlled trial”. This choice is due to the limited number of participants (n=17)

#KEYWORDS:

* Consider also adding: EMG, integrative and complementary medicine, balance,

#ABSTRACT

* methods: please declare if there was a blindness of the assessor, therapist, or patients.

* results: please not report only p-value. Add also the Treatment effect, mean (95% CI)

* FR: it is an abbreviation. Please report it in full.

#INTRODUCTION

* background: it lacks the essential details about the supposed effects of plantar reflexology. A gentle touch represents it. Therefore, it can work through different mechanisms (e.g., neurophysiology, biomechanics, placebo and contextual effects, body and space perception) shared with other manual therapy strategies (e.g., massage, joint mobilization). Accordingly, I suggest authors read and add the following fundamental papers concerning the proposed mechanisms of this type of gentle manual treatment (e.g., not only visceral influence). This strategy will enhance the value of the manuscript.

-Rossettini G, Latini TM, Palese A, Jack SM, Ristori D, Gonzatto S, Testa M. Determinants of patient satisfaction in outpatient musculoskeletal physiotherapy: a systematic, qualitative meta-summary, and meta-synthesis. Disabil Rehabil. 2020 Feb;42(4):460-472. doi: 10.1080/09638288.2018.1501102.

-Geri T, Viceconti A, Minacci M, Testa M, Rossettini G. Manual therapy: Exploiting the role of human touch. Musculoskelet Sci Pract. 2019 Dec;44:102044. doi: 10.1016/j.msksp.2019.07.008.

-Viceconti A, Camerone EM, Luzzi D, Pentassuglia D, Pardini M, Ristori D, Rossettini G, Gallace A, Longo MR, Testa M. Explicit and Implicit Own's Body and Space Perception in Painful Musculoskeletal Disorders and Rheumatic Diseases: A Systematic Scoping Review. Front Hum Neurosci. 2020 Apr 9;14:83. doi: 10.3389/fnhum.2020.00083.

-The mechanisms of manual therapy in the treatment of musculoskeletal pain: a comprehensive model.Bialosky JE, Bishop MD, Price DD, Robinson ME, George SZ.Man Ther. 2009 Oct;14(5):531-8. doi: 10.1016/j.math.2008.09.001

- Unraveling the Mechanisms of Manual Therapy: Modeling an Approach.Bialosky JE, Beneciuk JM, Bishop MD, Coronado RA, Penza CW, Simon CB, George SZ.J Orthop Sports Phys Ther. 2018 Jan;48(1):8-18. doi: 10.2519/jospt.2018.7476

*rationale of the study: I suggest authors report more clearly the existing study on plantar reflexology performed on patients with diabetes or other pathology. They also should show in detail the lack of literature to justify their treatments.

#MATERIAL & METHODS:

* reporting guidelines: the authors developed an RCT. However, they included a very limited number of participants (n=17). Moreover, they did not declare any followed guidelines in the text, but only in figure 2 (e.g., CONSORT). Accordingly, due to the nature of the study and the methodology adopted, I suggest that authors organize the trial as a “pilot controlled RCT” following the reporting of CONSORT guidelines for the pilot study, citing the appropriate literature. This strategy will improve the overall quality and clarity of the reporting.

Have a look here: http://www.consortstatement.org/extensions/overview/pilotandfeasibility

* Study design: report a statement where you declare that the study was performed following the Ethical principles of Helsinki with appropriate reference.

* Eligibility: ICF is an abbreviation. Report it in full. Moreover, report references for your inclusion and exclusion criteria.

* Procedures: please declare if there was a blindness of the assessor, therapist, or patients.

* Statistical analysis: please not report only the p-value. Add also the Treatment effect, mean (95% CI). Moreover, declare the statistical software adopted for the analysis (e.g., spss, r), the version and the year.

#RESULTS:

*please do not report only the p-value. Add also the Treatment effect, mean (95% CI).

#DISCUSSIONS

*” This finding could be explained by the touch produced 272 by FR [18]. Massage activates receptors for pressure, temperature, and nociception, as well 273 as increases local blood flow and muscle relaxation in the cutaneous area of the feet 274 [15,33]”. Once again, it would help if you discussed all the possible mechanisms induced by manual touch. Again, please consider adding a short discussion on the proposed references suggested in the introduction about the effects of the hands-on technique. This strategy will enrich your discussion.

*” The results found in the study by Silva et…. and Yümin et al.”. It would help if you discussed these studies, not only cited them. Please show me the similarities and differences.

* This result is justified according to Peppa 290 et al. [44], Peppa et al. [44]”. Here there is a repetition. Please, revise and rewrite this passage. Moreover, this justification is very well organized. Congratulation. Also, in other populations (e.g., fibromyalgia patients), insulin resistance was proposed as a small-fibre neuropathy mechanism. Consider adding and discussing this point well.

-Viceconti A, Geri T, De Luca S, Maselli F, Rossettini G, Sulli A, Schenone A, Testa M. Neuropathic pain and symptoms of potential small-fibre neuropathy in fibromyalgic patients: A national online survey.Joint Bone Spine. 2021 Jul;88(4):105153. doi: 10.1016/j.jbspin.2021.105153.

-Diabetic peripheral neuropathy: advances in diagnosis and strategies for screening and early intervention.Selvarajah D, Kar D, Khunti K, Davies MJ, Scott AR, Walker J, Tesfaye S.Lancet Diabetes Endocrinol. 2019 Dec;7(12):938-948. doi: 10.1016/S2213-8587(19)30081-6.

*limitations: you should modulate the conclusion showing the limit of your research. For example, you compared only treatment (specific) VS no treatment (control/wait and see). However, you missed the opportunity to add also another group (sham/placebo control). Without this type of comparison (specific VS placebo/sham), you cannot have any definitive conclusion on the treatment effect. Indeed, different contextual factors (e.g., the rituality of the treatment) presented during the administration of hands-on technique (e.g., plantar reflexology) can create a confounding effect between groups. Accordingly, I strongly suggest authors read and add this relevant paper that discusses the topic.

-Rossettini G, Testa M. Manual therapy RCTs: should we control placebo in placebo control? Eur J Phys Rehabil Med. 2018 Jun;54(3):500-501. DOI: 10.23736/S1973-9087.17.05024-9.

-Problematic placebos in physical therapy trials.Maddocks M, Kerry R, Turner A, Howick J.J Eval Clin Pract. 2016 Aug;22(4):598-602. doi: 10.1111/jep.12582

*implications: add a paragraph for the discussion on the implications for researchers and clinicians.

#CONCLUSION

* Following the suggested limitations, please modulate the results of your study.

Author Response

Reviewer's Suggestions 2

First I would like to thank you for the suggestions. The suggestions were very important for the improvement of the manuscript. Below I have described item by item in order of correction

#TITLE:

* I suggest changing the title, adding “: a pilot randomized controlled trial”. This choice is due to the limited number of participants (n=17)

Was inserted in the title

Effect of foot reflexology on muscle electrical activity, pressure, plantar distribution, and body sway in patients with type 2 diabetes mellitus: A pilot randomized controlled trial

#KEYWORDS:

* Consider also adding: EMG, integrative and complementary medicine, balance,

Keywords: type 2 diabetes mellitus; foot reflexology; rehabilitation, EMG, integrative and complementary medicine, balance.

#ABSTRACT

* methods: please declare if there was a blindness of the assessor, therapist, or patients.

* results: please not report only p-value. Add also the Treatment effect, mean (95% CI)

* FR: it is an abbreviation. Please report it in full.

Abstract: Objective: To verify the effect of foot reflexology on the electrical muscle activity of the lateral and medial gastrocnemius muscle, and to examine the distribution, plantar pressure, and body sway in patients with type 2 diabetes mellitus. Methods: This pilot randomized controlled trial enrolled 17 volunteers who were clinically diagnosed with diabetes mellitus. The sample was assigned to one of two groups: the control group (CG, n=7), who received information on foot care and health, and the intervention group (IG, n=10), who received the application of foot reflexology on specific areas of the feet, for 10 consecutive days. There was blinding of the evaluator and the therapist. Surface electromyography (EMG) was used to assess the electrical activity of the medial and lateral gastrocnemius muscles in maximum voluntary isometric contraction (MVIC) and isotonic contraction (IC); baropodometry and stabilometry were used to analyze unloading, plantar weight distribution, and body sway. Results: There was a statistically significant difference for the variables of maximum peak electrical activity of the left medial gastrocnemius (p=0.03; effect size=0.87 and power=0.81) and left lateral gastrocnemius muscles (p=0.04, effect size=0.70 and power=0.66) respectively, in the intragroup IC, and median frequency of the left medial gastrocnemius muscle in the intragroup MVIC (p=0.03; effect size=0.64 and power=0.59) , and in the variables intergroups of total area on the right side (p=0.04; effect size=1.03 and power=0.50) and forefoot area on the left side (p=0.02; effect size=0.51 and power=0.16).Conclusion:  We conclude that foot reflexology influenced some variables of intergroup plantar distribution and intragroup EMG in the sample studied. There is a need for a placebo group, a larger sample and a follow-up to strengthen the findings of these experiments

#INTRODUCTION

* background: it lacks the essential details about the supposed effects of plantar reflexology. A gentle touch represents it. Therefore, it can work through different mechanisms (e.g., neurophysiology, biomechanics, placebo and contextual effects, body and space perception) shared with other manual therapy strategies (e.g., massage, joint mobilization). Accordingly, I suggest authors read and add the following fundamental papers concerning the proposed mechanisms of this type of gentle manual treatment (e.g., not only visceral influence). This strategy will enhance the value of the manuscript.

-Rossettini G, Latini TM, Palese A, Jack SM, Ristori D, Gonzatto S, Testa M. Determinants of patient satisfaction in outpatient musculoskeletal physiotherapy: a systematic, qualitative meta-summary, and meta-synthesis. Disabil Rehabil. 2020 Feb;42(4):460-472. doi: 10.1080/09638288.2018.1501102.

-Geri T, Viceconti A, Minacci M, Testa M, Rossettini G. Manual therapy: Exploiting the role of human touch. Musculoskelet Sci Pract. 2019 Dec;44:102044. doi: 10.1016/j.msksp.2019.07.008.

-Viceconti A, Camerone EM, Luzzi D, Pentassuglia D, Pardini M, Ristori D, Rossettini G, Gallace A, Longo MR, Testa M. Explicit and Implicit Own's Body and Space Perception in Painful Musculoskeletal Disorders and Rheumatic Diseases: A Systematic Scoping Review. Front Hum Neurosci. 2020 Apr 9;14:83. doi: 10.3389/fnhum.2020.00083.

-The mechanisms of manual therapy in the treatment of musculoskeletal pain: a comprehensive model.Bialosky JE, Bishop MD, Price DD, Robinson ME, George SZ.Man Ther. 2009 Oct;14(5):531-8. doi: 10.1016/j.math.2008.09.001

- Unraveling the Mechanisms of Manual Therapy: Modeling an Approach.Bialosky JE, Beneciuk JM, Bishop MD, Coronado RA, Penza CW, Simon CB, George SZ.J Orthop Sports Phys Ther. 2018 Jan;48(1):8-18. doi: 10.2519/jospt.2018.7476

*rationale of the study: I suggest authors report more clearly the existing study on plantar reflexology performed on patients with diabetes or other pathology. They also should show in detail the lack of literature to justify their treatments.

…The application of manual therapy techniques such as massage, joint mobilization and myofascial release are capable of evoking a cascade of neurophysiological responses from the central and peripheral nervous system, biomechanics and psychological responses arising from somatosensory stimuli (Geri et al, 2019; Bialosky, 2009) . These stimuli modulate pain responses, affective and somatoperceptual responses observed in clinical outcomes [17,18]. (Geri et al, 2019; Bialosky,2009).

Meta-analytic studies and systematic reviews report that foot reflexology is a promising method for improving sleep, fatigue, pain, mood, nausea, and quality of life in several populations [19]. When foot reflex massage is applied to patients with diabetic neuropathy, it is able to reduce their complaints [20]. (Agustini et.al., 2019), modify the level of hemoglobins and increase peripheral vascular circulation [21]. (Yodsirajinda et. al., 2016) and improve general fatigue, heart rate and mood [22]. (Kim, 2003).

However, these studies show low quality of scientific evidence, low methodological quality and few randomized clinical trials. To date, there are no randomized clinical trial studies that analyzed the pattern of muscle activation and parameters of plantar pressures in the population with type 2 diabetic neuropathy that applied foot reflexology.

#MATERIAL & METHODS:

* reporting guidelines: the authors developed an RCT. However, they included a very limited number of participants (n=17). Moreover, they did not declare any followed guidelines in the text, but only in figure 2 (e.g., CONSORT). Accordingly, due to the nature of the study and the methodology adopted, I suggest that authors organize the trial as a “pilot controlled RCT” following the reporting of CONSORT guidelines for the pilot study, citing the appropriate literature. This strategy will improve the overall quality and clarity of the reporting.

Have a look here: http://www.consortstatement.org/extensions/overview/pilotandfeasibility

* Study design: report a statement where you declare that the study was performed following the Ethical principles of Helsinki with appropriate reference.

2.1. Study design

This pilot randomized controlled clinical trial recruited participants between September 2019 and February 2020. All volunteers were informed about the objectives of the study, the importance of the activities developed, and the possible outcomes. Additionally, they voluntarily signed an informed consent form, according to the determinations of Resolution 466/12. This study was approved by the Research Ethics Committee of the Faculdade de Ciências Médicas (nº. 659.819) and was conducted in accordance with the Declaration of Helsinki [23]. This study obtained the clinical trial under the number RBR-7wjrkp9. This study followed the guidelines of the randomized clinical trial studies for the pilot study.

* Eligibility: ICF is an abbreviation. Report it in full. Moreover, report references for your inclusion and exclusion criteria.

2.3. Eligibility criteria

Volunteers of both genders with T2DM, aged over 40 years, with 3 years or more of diabetes onset, a visual analog pain scale with a moderate level (i.e., from mark 4 on the scale) [24], and who agreed to participate in the study and informed consent were included in the study.

* Procedures: please declare if there was a blindness of the assessor, therapist, or patients.

2.4. Sample randomization

Randomization was performed by a blinded researcher (Researcher 1). The names of the volunteers were numbered and placed in a brown, sealed envelope. Researcher 2 opened the envelope and performed the draw, in which he/she allocated participants to one of two groups: the control group (CG) did not perform the intervention) and the intervention group (IG) performed the foot reflexology massage). The researcher who evaluated and the researcher who performed an intervention were blinded.

* Statistical analysis: please not report only the p-value. Add also the Treatment effect, mean (95% CI). Moreover, declare the statistical software adopted for the analysis (e.g., spss, r), the version and the year.

#RESULTS:

*please do not report only the p-value. Add also the Treatment effect, mean (95% CI).

2.10. Statistical analysis

The descriptive analysis of data was used to characterize the sample by mean and standard deviation for continuous variables and percentages for categorical variables. The Shapiro–Wilk test was applied to determine normality of the data, and independent t-tests were applied for the variables age, weight, height, BMI, shoes, time of diabetes, visual analog scale, EMG, distribution and plantar pressure, and body sway; whereas the Chi-square test was applied for the variables gender, DN, culturing sensitivity, neuropathic symptoms, and neuropathic impairment. Paired t-tests were used for the variables EMG, plantar distribution and pressure, and body sway. All analyzes were performed by the SPSS statistical software (version 20.0).

… Table 2 shows a statistically significant intragroup difference in the median frequency of the left medial gastrocnemius (p=0.03) and intergroup there was no statistically significant difference in the variables analyzed (p>0.05). The mean difference for the median frequency variable for the left GM was 144.65 post-intervention, the confidence interval was between 104.32 a 184.97. Note that the confidence interval values ​​did not pass through 0, indicating effect produced by the technique in relation to pre-intervention and post-intervention. The effect size was 0.64, indicating a medium effect size. The power was 0.59, indicating low power.

… In Table 3, it is noted that there was a statistically significant difference within the intragroup in the variables maximum peak of the left lateral gastrocnemius and left medial gastrocnemius (p=0.04 and p=0.03, respectively), and there was no intergroup difference in the variables analyzed (p>0.05).

The mean of the variable maximum peak of the left lateral gastrocnemius was 82.55 after the intervention, the confidence interval was between 72.26 and 92.84. The mean of the maximum peak variable of the left medial gastrocnemius was 83.62 post-intervention, the confidence interval was between 75.53 and 91.72. Note that the confidence interval did not pass the value 0, indicating an effect of the technique in relation to pre-intervention and post-intervention. The effect size for a left lateral gastrocnemius maximum peak variable was 0.70, indicating a mean effect size. The power was 0.66, indicating low power. The effect size for a maximal left medial gastrocnemius peak variable was 0.87, indicating a high effect size. The power was 0.81, indicating high power.

… Table 4 presents the mean, standard deviation, and 95% confidence interval of the distribution and plantar pressure variables, for CG and GI. A statistically significant difference is seen for the total surface area of the left foot and surface area of the left forefoot intergroup variables (p>0.05). We then determined the differences in the means between groups for total surface area of the left foot variable (17.24) with confidence interval (0.44 to 34.04) and surface area of the left forefoot variable (7.65) with confidence interval (–7.68 to 2.28). Note that the confidence interval values did not pass through 0, indicating effect produced by the technique for the variable total surface area of the left. The effect size for the total surface area of the left foot variable was 1.03, indicating was a high effect size. The power was 0.50, indicating low power. The effect size for the surface area of the left forefoot variable was 0.51, indicating was a low effect size. The power was 0.16, indicating low power.

#DISCUSSIONS

*” This finding could be explained by the touch produced 272 by FR [18]. Massage activates receptors for pressure, temperature, and nociception, as well 273 as increases local blood flow and muscle relaxation in the cutaneous area of the feet 274 [15,33]”. Once again, it would help if you discussed all the possible mechanisms induced by manual touch. Again, please consider adding a short discussion on the proposed references suggested in the introduction about the effects of the hands-on technique. This strategy will enrich your discussion.

… Studies indicate that massage is capable of producing pain modulation, reducing muscle spasms and joint stiffness [18,30]. The tactile information evoked by massage stimulates the fast and large-caliber nerve fibers (Aβ and Aδ) inhibiting the slow, small-caliber fibers, consequently reducing the perception of pain [18], as well as increasing the release of neurotransmitters and thus, playing an important role in modulation [30]. Massage-induced mechanical pressure generated changes in muscle-tendon compliance by mobilizing and stretching connective cells, improving joint and muscle stiffness [30].

The act of touching the surface of the skin evokes deactivation of the system related to the stress and threat response, that is, generating affective effects. Massage can also improve body perceptions through the reorganization of mental representations of the body [18].

*” The results found in the study by Silva et…. and Yümin et al.”. It would help if you discussed these studies, not only cited them. Please show me the similarities and differences.

The findings of Megda et. al. [33], agree with the findings of the present study, who verified the immediate effect of reflexology on the variable mass division in the plantar area with eyes closed in patients with diabetic neuropathy. Yümin et al. [32] obtained positive responses in relation to mobility, balance and functional range after the application of foot massage in patients with diabetic neuropathy, corroborating the findings of the present study.

* This result is justified according to Peppa 290 et al. [44], Peppa et al. [44]”. Here there is a repetition. Please, revise and rewrite this passage. Moreover, this justification is very well organized. Congratulation. Also, in other populations (e.g., fibromyalgia patients), insulin resistance was proposed as a small-fibre neuropathy mechanism. Consider adding and discussing this point well.

… The results found in the muscle electrical activity variables denote a lower muscle activation in isotonic and isometric contractions. This result can be justified, due to the fact that the muscular components and the neural components of the motor units in patients with T2DM are altered both in their structure and in functionality, these alterations may be associated with insulin resistance Peppa et al [34].

*limitations: you should modulate the conclusion showing the limit of your research. For example, you compared only treatment (specific) VS no treatment (control/wait and see). However, you missed the opportunity to add also another group (sham/placebo control). Without this type of comparison (specific VS placebo/sham), you cannot have any definitive conclusion on the treatment effect. Indeed, different contextual factors (e.g., the rituality of the treatment) presented during the administration of hands-on technique (e.g., plantar reflexology) can create a confounding effect between groups. Accordingly, I strongly suggest authors read and add this relevant paper that discusses the topic.

*implications: add a paragraph for the discussion on the implications for researchers and clinicians.

…The present study has the following limitations: the need for a placebo group, as the ability to explore the findings and draw conclusions was limited; the sample size was smaller than necessary to obtain statistical significance of the results, although the results indicate clear trends for the intervention group. Future work to increase the sample size to obtain statistically significant results and follow-up from participants after longer periods of time would strengthen the findings of these experiments.

The clinical implications in the present study are being pointed out as a low-cost technique that is easily applied by professionals in the public health network.

#CONCLUSION

* Following the suggested limitations, please modulate the results of your study.

… We conclude that foot reflexology influenced some variables of intergroup plantar distribution and intragroup EMG in the sample studied. There is a need for a placebo group, a larger sample and a follow-up to strengthen the findings of these experiments

Round 2

Reviewer 2 Report

Dear authors

Congratulations: you have improved your work!

It is ready to be accepted.

Best regards.